# Analysis of Main Components of Five Mulberry Varieties in Tropics

**DOI:** 10.3390/plants13192763

**Published:** 2024-10-01

**Authors:** Dezhao Lou, Huazhou Wu, Hongxian Wei, Fuping Lu, Tao Geng, Peiqun Lin, Shuchang Wang

**Affiliations:** 1National Key Laboratory for Tropical Crop Breeding, Sanya 572024, China; loudezhao@163.com (D.L.); wuhuazhouaa@126.com (H.W.); hongxianw0816@163.com (H.W.); fuping_36@163.com (F.L.); wstqgt@126.com (T.G.); linpeiqun520@163.com (P.L.); 2Key Laboratory of Integrated Pest Management on Tropical Crops Ministry of Agriculture and Rural, Hainan Key Laboratory for Monitoring and Control of Tropical Agricultural Pests, Institute of Environment and Plant Protection, Chinese Academy of Tropical Agricultural Sciences, Haikou 571101, China; 3College of Tropical Crops, Hainan University, Haikou 570228, China

**Keywords:** morphology, sugar, organic acid, amino acid, aroma contents

## Abstract

Mulberries (*Morus alba* L.) contain rich and beneficial nutrients for human health. However, as a temperate adaptive species, high-temperature and high-humidity climate conditions may alter the main nutritional value of mulberries after their intended arrival in tropical regions, which has not yet been reported on. In this study, we analyzed the differences in morphology, sugars, organic acids, free amino acids, and aroma contents of five mulberry varieties in the tropics between two harvesting periods. The results show that the full-ripe fruits of *M. laevigata* W (TLM) have the longest fruit length (83.67 mm) and highest brix (25.90); meanwhile, full-ripe fruits of *M. atropurpurea* R (D10M) have the longest fruit transverse stem (20.00 ± 0.577 mm) and single-fruit weight (9.63 ± 0.033 g). Fructose, glucose, and sucrose were the main sugars, and oxalic acid, quinic acid, malic acid, and citric acid were the main organic acids in all varieties; in addition, the sucrose content in mature fruits of *M. laevigata* W. (BLM) and *M. alba* L. BZZ (BZM) was significantly higher than other sugars. Twenty free amino acids were detected in all five varieties and asparagine was the main free amino acid. A total of 100 volatile compounds were identified, including 31 esters, 20 aldehydes, 14 hydrocarbons, 11 alcohols, 10 acids, 6 ketones, and 8 others. Although the main components of five mulberry full-ripe fruits were significantly higher than the green-ripe fruits, gamma-amino butyric acid and a few other components were otherwise. The research results show that the tropical climate conditions could increase the main nutritional components of mulberries, but the specific molecular regulatory mechanisms need to be further analyzed.

## 1. Introduction

Mulberry is an important component of sericulture, and the fruit is deeply loved by people because of its delicacy and nutritional value [1]. Mulberries are not only rich in nutrients essential for the human body, but also contain medicinal active ingredients such as flavonoids and alkaloids, making them one of the agricultural products that are both food and medicine [2]. The mulberry fruit contains appropriate amounts of soluble solids, flavonoids, polyphenols, polysaccharides, volatile oils, amino acids, vitamins, and trace elements [3]. Therefore, the long-term consumption of mulberries is beneficial for human health.

The fruit of mulberry has always been used as a food or beverage, but we lack an awareness of its use as medicine in the past, due to the above specific issues, but this use is gradually being more widely studied at present. Frozen mulberry fruit powder improves blood and liver antioxidant status and improves eyesight in rats [4]. Mulberry has also been used to treat obesity, and for premature hair whitening [5]. Different types of anthocyanins have been isolated from mulberry fruit [6,7]. The fruits of *Morus alba* L. and *M. rubra* L. are abundant in polyphenolic compounds, such as quercetin, rutin, and 1-deoxynojirimycin, which impede some obesity-induced illnesses [8]. Kim et al. isolated a variety of pyrrole alkaloids from mulberry that can inhibit the activity of pancreatic lipase and can be used for weight loss [9,10]. Research has shown that mulberry is rich in polyphenols and exhibits antioxidant activities [11]. Cyanidin-3-O-β-D-glucopyranoside (C3G) has been extracted from mulberry fruit, which is beneficial for endothelial dysfunction [12,13,14]. Essential amino acids and minerals in the human body were detected in the Chinese mulberry cultivar “Hetianbaisang” [15].

The change in cultivation environment not only affects the yield and growth rate of crops, but also directly affects the quality. At present, there are few reports on the analysis of the nutritional value of mulberry fruits in tropical regions. Therefore, we screened suitable mulberry varieties for cultivation in tropical regions, analyzed the nutritional components of mulberry at different ripeness stages, and comprehensively evaluated its nutritional value. The research results will provide theoretical guidance for the selection of mulberry varieties for different purposes in tropical regions, and will also provide guidance for the breeding of mulberry varieties in tropical regions. This study aimed to determine the morphological and chemical characteristics of fruits of five mulberry varieties with two ripeness stages from Danzhou County, Hainan Province, a tropical region in China. High-performance liquid chromatography (HPLC) was used to determine the composition and content of the organic acids, sugars, and amino acids. Solid-phase microextraction (SPME) and gas chromatography–mass spectrometry (GC-MS) were used to analyze the volatile aroma components. The purpose is to comprehensively evaluate the quality and medicinal value of mulberry fruit, and provide theoretical bases for the comprehensive utilization of mulberry fruit.

## 2. Results

### 2.1. Variation in Morphology and Brix of Different Mulberry Varieties and Ripeness Stages

We compared and analyzed the morphological characteristics of five mulberry fruit varieties suitable for cultivation in tropical regions, as shown in Figure 1.

The morphological and sweetness data of the two mature mulberry varieties were analyzed by variance analysis, using two-way ANOVA, which revealed that the varieties had significant effects on fruit length (*F* = 858.266, *p* = 0.000), transverse stem length (*F* = 166.238, *p* = 0.000), fruit weight (*F* = 3246.023, *p* = 0.000), and brix (*F* = 171.833, *p* = 0.000). Fruit ripeness stages significantly affected fruit weight (*F* = 36.364, *p* = 0.000), fruit width (*F* = 9.333, *p* = 0.006), and brix (*F* = 640.774, *p* = 0.000). The interaction between variety and maturity had significant effects on single fruit weight (*F* = 3.864, *p* = 0.017) and brix (*F* = 25.183, *p* = 0.000); however, the interaction between variety and maturity had no significant effect on the fruit length (*F* = 0.266, *p* = 0.896) or fruit transverse stem (*F* = 0.286, *p* = 0.884) of mulberry, as shown in Table 1.

In all mulberry varieties, the full-ripe TLM mulberry was the longest, measuring 83.67 mm, the green-ripe BZM was the shortest, measuring 23.67 mm. The full-ripe D10M mulberry had the largest fruit transverse stem, measuring 20.00 mm, whereas the green-ripe BZM mulberry had the smallest, measuring 7.33 mm. The weight of a single fruit green-ripe BZM (2.27 g) was the lowest, whereas the weight of a single full-ripe D10M (9.63 g) was the highest, and both were significantly different from the other varieties. We analyzed the brixes of two different-maturity mulberry varieties and found that the brix of the long fruit TLM was the highest, followed by BLM and BZM (Table 1).

### 2.2. Sugar and Organic Acid Contents of Different Mulberry Varieties and Ripeness Stages

Three different sugars, fructose, glucose, and sucrose, of five diverse mulberry varieties at two ripeness stages, were measured by HPLC. It was found that both variety and maturity significantly affected fructose, glucose and sucrose contents.

The concentrations of individual soluble sugars in the mulberry varieties during fruit ripening are shown in Figure 2A. The concentrations of fructose, glucose and sucrose generally increase during ripening. The trends of fructose and glucose content in different varieties of ripe fruits were consistent, with those contents increasing from 12.66 to 62.77 mg/g FW and 11.78 to 61.19 mg/g FW, respectively. The concentrations of fructose and glucose in full-ripe JL40M were significantly higher than in other varieties, reaching 62.77 mg/g FW and 61.19 mg/g FW, respectively. In contrast, a significantly lower level of fructose was found in green-ripe BLM, with only 12.66 ± 0.98 mg/g FW and 11.78 mg/g FW. The concentration of sucrose in BZM was the highest, reaching 99.17 nmg/g FW, which was 1.68 times higher than the second-highest concentration in BLM, and 15.7 times higher than the lowest concentration in JL40M. In the green-ripe mulberry, the changes in fructose and glucose concentrations were consistent with those in the ripe mulberry. However, sucrose concentration in the green-ripe fruit was higher than that in the full-ripe fruit JL40M.

Regarding the ratio of sugars (Figure 2B), the ratio of fructose in green-ripe mulberry varieties was the highest, ranging from 37% to 51%, followed by glucose, where it accounted for 36% to 46%. However, in the full-ripe mulberry, the ratios of individual soluble sugars varied among the different varieties. Sucrose accounted for the largest proportion of full-ripe BZM and BLM, with 54% and 46%, respectively, whereas the highest ratio of fructose was found in full-ripe JL40M, D10M, and TLM, with 48%, 47%, and 40%, respectively.

Oxalic acid, quinic acid, malic acid and citric acid were detected in different mulberry varieties at two ripeness stages, and a two-way ANOVA showed that the variety significantly influenced oxalic acid, quinic acid, malic acid and citric acid, whereas ripeness stages had an obvious impact on oxalic acid and citric acid, as is shown in Table A2.

The ratios of organic acids (Figure 3B) and the changes in organic acid composition and concentration were observed in different mulberry varieties at two ripeness stages. Citric acid was the predominant organic acid in BLM, TLM, D10M, and JL40M during the green-ripe period. In BZM, oxalic acid was the predominant organic acid during the green-ripe period, whereas oxalic acid and malic acid were the predominant organic acids during the full-ripe period. The contents of the other types of organic acids varied among the different varieties. The concentration of oxalic acid gradually decreased during mulberry ripeness stages. The highest concentration of oxalic acid was found in the green-ripe BZM, with an approximately 4-fold difference compared with the green-ripe BLM. This indicates that quinate also follows a similar decreasing trend, but the highest concentration was found in the BLM variety, while the malic acid concentration was high in the TLM, BZM, and JL40M varieties at the full-ripe stage. The citric acid concentration decreased during the fruit-ripening process in the TLM and D10M varieties, whereas the other varieties showed only a slight change in concentration.

### 2.3. Free Amino Acid Content of Different Mulberry Varieties and Ripeness Stages

Twenty free amino acids were identified and quantified in different mulberry varieties at two ripeness stages. These included nine essential amino acids (Thr, Val, Met, Ile, Leu, Phe, Trp, Lys, and His), with Met present only in trace amounts. The remaining 12 amino acids analyzed were Asp, Glu, Asn, Ser, Gln, Gly, Gamma-aminobutyric acid (GABA), Arg, Ala, Pro and Tyr. Two-way ANOVA showed that variety significantly influenced all 20 free amino acids, whereas ripeness stages had a noticeable impact on 18 free amino acids, except for Tyr and Met. This is shown in Table A4.

The sum of all amino acids in each sample ranged from 225.72 to 651.3 mg/100 g; the highest concentration was discovered in full-ripe BZM, with approximately a 5.4-fold difference from green-ripe TLM (shown in Table A3).

The proportions of amino acid content varied among different varieties. Except for the D10M variety, wherein the proportion decreased with increasing ripeness stages, all other varieties showed an increase with increasing ripeness stages, with JM40.S1 having the highest proportion of 61.4% sample content. The highest proportion of Glu content was 21.07% in the BLM.S2 sample, and the contents of all varieties were higher in the S2 stage than in the S1 stage. The change in GABA content showed the opposite trend, with the D10M.S1 sample reaching the highest proportion (Figure 4A).

We compared and analyzed the correlations between amino acids based on their contents, as shown in Figure 4B. Among the seven essential amino acids identified, the levels of Thr, Val, Met, Leu, Phe, Trp, and Lys in the full-ripe BZM were significantly higher than in other samples, and only BZM had the necessary amino acid content, which increased during the process of maturity. Additionally, except for Gly, 11 other amino acids were present in high concentrations. Interestingly, in addition to glutamic acid and proline, the contents of nine other amino acids were negatively correlated with mulberry maturity in the BLM samples. The concentration of GABA was high in mature mulberry.

### 2.4. Aroma Components of Different Mulberry Varieties and Ripeness

A total of 100 volatile compounds with a range of 81~95 unique compounds were identified in ten samples. Based on their functional groups, the identified volatiles were classified into seven groups: 31esters, 20 aldehydes, 14 hydrocarbons, 11 alcohols, 10 acids, 6 ketones, and 8 others. As shown in Figure 5A, the most abundant volatile compounds were aldehydes, accounting for 40.83–74.75% of the total volatiles in the samples, followed by esters (7.41–36.32%) and alcohols (2.40–29.62%). The remaining other groups of volatile compounds accounted for only1.52–4.89% of the total volatiles. This is shown in Table A5.

The aroma components of all varieties decreased as the ripeness stages proceeded. Regarding the aroma components, green-ripe D10M had the greatest number (85 species). The aroma components of the full-ripe BLM were the best, with only 76 species. The number of common aroma component species in the two ripeness stages of the five mulberry varieties was 57.

The contents of aldehydes were the highest not only between different mulberry varieties but also between the two ripeness stages, and the aldehyde content gradually increased with increasing maturity, while the content of green-ripe BLM was higher than that of the others. The content of 2-Hexenal, (E)-, was the highest among those aldehydes, and was highest in green-ripe BLM (47.35%), whereas it was lowest in the full-ripe BLM (27.69%). The second highest aldehyde content was seen for hexanal, which was abundant in green-ripe and full-ripe BZM and JL40M, and full-ripe D10M, at 19.14%, 18.39%, 18.12%, and 17.87%, respectively.

The content of esters in mulberry fruits was usually higher in the green-ripe stage than in the full-ripe one, but in the BLM variety it was higher in the full-ripe stage, reaching up to 56.19%. The varieties contained different ester substances; the main substance in BLM was 2(3H)-Furanone 5-hexyl-dihydro-, present at 8.78% in green-ripe but 21.80% in full-ripe fruits. 2-Hexen-1-ol, acetate, and (E)- were the most abundant ester substances in green-ripe TLM, with a content of 8.94%, but the content of the main substances 2-Hexen-1-ol, acetate, and (E)- in the TLM.S2 sample were 5.14%. Hexyl butyrate was the most abundant in the green-ripe D10M (4.76%), whereas hexyl acetate was the most abundant substance in the full-ripe D10M (4.83%) and BLM(3.49%).

There were no significant changes in alcohol content in both ripeness stages of TLM, BLM, and D10M. However, the alcohol content decreased during the ripening process for JL40M and BZM. The concentration of 2-Hexen-1-ol, (E)-, was the highest in TLM and BLM, whereas 1-Hexanol had the highest concentration in JL40M, D10M, and BZM. The acids, hydrocarbons, ketones, and other substance contents were comparatively low.

To visualize and compare the individual volatiles in the five mulberry varieties at the two ripeness stages, a heat map cluster analysis was conducted (Figure 5B). Clustering based on the volatile substance content at the two ripeness stages of different mulberry varieties clearly distinguished the full-ripe BLM. The full-ripe BLM sample showed negative correlations with almost all the selected volatiles (black color in Figure 5B), but it was positively correlated with volatile compounds such as 2-hexenal, 2(3H)-furanone, hexanal, 1-hexanol, acetic acid, 2-hexen-1-ol, and phenylethyl.

Further analysis clustered the five mulberry varieties into two ripeness stage groups. Compared with the other samples, BLM at the two ripeness stages showed a stronger positive correlation with volatiles derived from 2-hexenal, hexanal, acetic acid, benzene acetaldehyde, and heptanal. 2-Hexenal, (E)-, had a higher content than other substances in all samples. The contents of 2 (3H)-Frunone, 5-hydroxyldihydro, and 8-Heptadecene increased during BLM ripening but they remained as they were at lower ripeness stages in other varieties, which may be the key to BLM’s unique aroma in later ripeness stages. Both n-Hexadecanoic acid and 2-Hexenoic acid showed an increasing trend during BLM ripening and were significantly more abundant than other varieties. Although 3,4-Dimethoxytoluene has a lower content in BLM, as a unique substance, it may play a key role in aroma formation. The heat map differentiated samples based on the volatile contents of different mulberry varieties at two stages of ripeness.

## 3. Discussion

In this study, excluding morphology and brix, a total of three sugars, four organic acids, 20 free amino acids, and 100 volatile compounds were identified in the two ripeness stages of five mulberry varieties.

The experimental results show that different mulberry varieties influenced the chemical content, and our findings are consistent with those of previous studies [16]. The species also influenced morphology, with TLM being the longest, followed by D10M, BLM, JL40M, and BZM. Compared with the green-ripe fruits, all brix numbers showed that S2 had a higher value than S1. This indicates that the full-ripe fruits of all varieties were sweeter. The same result was observed in *Morus* laevigata Wall, where the fruits that ripened later were much sweeter [17].

The two ripeness stages of the five mulberry varieties contained fructose, glucose, sucrose, and other sugars. Fructose, glucose, and sucrose have also been detected in other mulberry varieties such as white (*M. alba* L.) and black (*M. nigra* L.) mulberry fruits, as well as in *M. rubra* L. In our experiment, the main components in JL40M, D10M, BLM, and TLM were fructose and glucose. Sucrose is rarely found in mulberry fruits, especially during the green-ripe period [18,19,20,21]. Mikulic-Petkovsek et al. suggested that fructose concentration is much more important in making fruits sweeter for consumers. However, in our experiment, TLM had the highest brix value, but the fructose content was relatively low. Therefore, the conclusion of Mikulic-Petkovsek et al. is controversial [22]. In our experiment, sucrose was the predominant substance of all sugars of BZM, reaching 99.17 mg/g FW, which is 1.68 times higher than the second-highest BLM and 15.7 times higher than the lowest JL40M. This high concentration can be attributed to the accumulation of sucrose in full-ripe fruits, which accounted for more than half of the total.

Oxalic acid, quinic acid, malic acid, and citric acid were detected in the two ripeness stages of the five mulberry varieties. Citric acid was the main component of organic acids in BLM, TLM, D10M, and JL40M. Similar results have been observed for *M. nigra*, *M. rubra*, and other *Morus* spp. fruits [23,24]. Oxalic acid and malic acid were the major organic acids in BZM; however, previous research has indicated that malic acid is the major component of mulberry fruits [25,26,27,28]. Therefore, in these experiments, the main types and proportions of organic acids varied among different mulberry species.

The amino acids of mulberry are transformed into aqueous proteins to form silk, which is the major product of sericulture [29,30]. A total of 20 free amino acids were detected in our experiment, including nine essential amino acids (Thr, Val, Met, Ile, Leu, Phe, Trp, Lys, and His), whereas Met only had trace levels, and the remaining 12 amino acids Asp, Glu, Asn, Ser, Gln, Gly, GABA (γ- aminobutyric acid), Arg, Ala, Pro, Tyr, and Cys were analyzed. GABA has been found in other mulberry leaves to inhibit angiotensin I-converting enzyme (ACE) activity in vitro, and has an antihypertensive effect [31,32]. In the full-ripe fruit of TLM and BZM and green-ripe fruit of BLM, GABA content is the highest, so we suggested that TLM, BZM and BLM can be used to collect and enrich GABA, and will be precious medicinal materials used to reduce high blood pressure in the medical industry.

The aroma substances of fruits are the main factors that constitute and affect the quality of fresh fruits and processed products [33]. Scholars have studied the flavor compounds of dried mulberries and identified 28 volatile compounds [34], suggesting that aldehydes are the main components of the flavor compounds in dried mulberries. We have identified a total of 100 volatile compounds, with aldehydes being the most abundant, which is consistent with our results. Terpene alcohols are characteristic aroma components of fruits. There are 11 identified terpene alcohol compounds in this experiment, mainly including leaf alcohol, n-hexanol, linalool, benzyl alcohol, etc. [35]. The proportion of terpene alcohols varies among different mulberry varieties, which may lead to different mulberry aromas. This study indicates that mulberries are rich in aroma compounds, including characteristic aroma components such as linalool and lactones, as well as other terpenoid alcohols and ketone compounds. Whether these components are characteristic aroma components of mulberries in tropical regions remains to be further analyzed and identified.

## 4. Materials and Methods

### 4.1. Plant Material

Here, 3-year-old mulberry trees were planted at the Chinese Academy of Tropical Agricultural Sciences Environment and Plant Protection Institute Mulberry Resource Garden, with the same cultivation measures. The climate is a tropical humid monsoon climate, with a minimum temperature of over 10 °C and a maximum temperature of 35 °C throughout the year, with an average temperature of over 24 °C. The annual rainfall is over 1500 mm. The five mulberry varieties were *M. laevigata* W (BLM), *M. laevigata* W (TLM), *M. atropurpurea* R (D10M), *M. alba* L Jialing (JL40M) and *M. alba* L BZZ (BZM). The S1 stage is the green-ripe stage, and the S2 stage is the full-ripe matured stage that is reached after 12 days of harvesting in S1. Here, 50 g of full-ripe or green-ripe fruits without any physical damage or fungal infections were collected, crushed and blended in a glass container at 100 rpm for five min, and we then took the supernatant for analysis of the main components immediately. In addition, 3 mulberry trees were randomly selected, and 10 mulberry varieties were selected from each tree for measuring fruit length, stem length, fruit weight, and weight. We took the average value of the measurement as one repetition, and repeated each treatment three times.

### 4.2. Extraction, Purification, and Isolation of Sugars and Organic Acids

Mulberry must (0.5 g) was used for the extraction (BS150M, Shanghai Yousheng Weighing Instrument Co., Ltd., Shanghai, China). Mulberry samples were pooled and ground into a fine homogenate using a mortar and pestle. The homogenate was diluted to 25 mL with redistilled water, ultrasonically extracted at 90 W (1006, Haoshun Ultrasonic Equipment, Shenzhen, China) for 30 min, cooled to room temperature, and centrifuged at 10,000× *g* for 10 min (5810R, Eppendorf, Hamburg, Germany). Thereafter, the supernatant was transferred to a 50 mL volumetric flask, the remaining samples were diluted to 15 mL with redistilled water and ultrasonic extraction was carried out at 90 W for 20 min. The sample was then centrifuged at 10,000× *g* for 10 min and cooled to room temperature. When combined with the supernatant of ultrasonic extraction, the volume was determined to be 50 mL after passing through a 0.45 µm Millipore filter (13 mm, 0.22 μm, Tianjin Jinteng Experimental Equipment Co., Ltd., Tianjin China). An aliquot (10 mL) of the supernatant was used for HPLC analysis. The sugar (fructose, glucose, and sucrose) and organic acid (oxalic acid, quinic acid, malic acid, and citric acid) contents were analyzed using HPLC (ACQUITY UPLC H-C1ass, Waters, Camden, NJ, USA). Soluble sugars were separated using a Sugar Pak TM I column (Waters, 4.6 mm × 250 mm, 5 µm) operated at 80 °C, with an injection volume of 10 μL. The mobile phase was bi-distilled water with a flow rate of 0.6 mL/min. The total run time was 25 min, and an RI detector was used to monitor soluble sugars. Organic acids were analyzed by HPLC using an Agilent HC-C18 exclusion column (4.6 mm × 250 mm, 5 µm) (ACQUITY UPLC H-C1ass, Waters, Camden, NJ, USA) associated with a PDA detector set at 245 nm. The column temperature was set at 50 ℃, with an injection volume of 10 μL. The elution solvent was 0.01 mM sulfuric acid in bidistilled water, at a flow rate of 0.5 mL/min. The duration of the analysis was 30 min. Sugars and organic acids are expressed as mg/g FW.

### 4.3. HPLC Analysis Method for Amino Acid Content

Extraction of amino acid samples: Take 0.5 g of sample powder, add 2.5 mL of 0.1 mol/L HCL to a 10 mL centrifuge tube, shake well, and place in sand ice. Put the ice water mixture into the ultrasonic instrument, place the centrifuge tube containing the sample powder and the rack together into the ultrasonic instrument, extract with ultrasound for 20 min, and shake once in the middle. Centrifuge at 10000 R for 10 min at 4 °C and collect the supernatant. Repeat the extraction once and combine the two supernatants. Transfer 300 µL of the supernatant into a 2 mL centrifuge tube. Add 75 µL of Derivative 1 solution and 75 µL of Derivative 2 solution, vortex for 1 min. Soak in water at 25 °C for 60 min. Add 1.4 mL of n-hexane, vortex for 15 s, and let it stand for 2 min. Centrifuge at 10,000 R for 5 min at high speed. Extract twice and use a 1 mL syringe to aspirate the lower-layer liquid. Filter the solution through a 0.22 µm membrane and transfer it to a chromatographic injection bottle for sample analysis.

Amino acids were analyzed by HPLC using an Agilent HC-C18 exclusion column (4.6 mm × 250 mm, 5 µm) associated with a PDA detector set at 254 nm. Mobile phase: A is a 0.05 mol/L sodium acetate trihydrate (pH 6.50), B is an acetonitrile aqueous solution with methanol:water = 6:2 (volume ratio) and flow rate of 1 mL/min. The column temperature was set at 35 °C, with an injection volume of 10 μL. The duration of the analysis was 40 min.

### 4.4. HS-SP ME-GCMS Analysis Method

The SPME fibers were conditioned at 250 °C for 60 min before use. Firstly, 0.50 g measures of ten ground samples were weighed into a 20 mL glass vial with 5 mL saturated salt solutions of NaCl (Sangon Biotech, Shanghai, China). Thereafter, the SPME fiber was pushed into the headspace of the vial to extract volatile compounds at 40 °C for 10 min. The analytes were finally desorbed for 2 min at 250 °C in a gas chromatograph injector in the splitless mode.

GC-MS analysis was performed on an Agilent 6890N-5973 GC-MS instrument (Agilent Technologies Inc., Santa Clara, CA, USA) with an Agilent 19091S-433 DB-5MS capillary column (30 m × 250 µm × 0.25 µm, Agilent Technologies Inc., Santa Clara, CA, USA) following the method described by Bi et al. [32] The initial oven temperature was 40 °C, which was maintained for 1 min, increased at 3.5 °C/min to 100 °C, and then increased to 180 °C at 4 °C/min. Finally, the temperature was increased to 250 °C at a rate of 5 °C/min and maintained for 3 min. Pure helium (99.999% purity) was used as the carrier gas at a flow rate of 1 mL/min. A mass detector was used in electron impact mode at 70 eV, and the ion source temperature was 230 °C. The mass spectra were scanned from 35 to 335 atomic mass unit. The volatile components were tentatively identified by comparing the mass spectra and RI in the standard NIST 08 library, and the internal standard method was used to calculate the levels of volatile compounds.

### 4.5. Statistical Analysis

Statistical analysis was conducted using Statistical Product and Service Solutions (SPSS) 19.0 (SPSS, Chicago, IL, USA). The normality of the distribution (original data or transferred data) was checked using the test of Homogeneity of Variances. One-way analysis of variance (ANOVA) was followed by Duncan test. The data differed from a normal distribution, and the statistical significance and level were checked with Nonparametric tests followed by Kruskal–Wallis one-way ANOVA. We performed two-factor analysis of variance using SPSS19.0. We drew a bar chart using Origin 2022 software and created a heatmap using the Package:Pheatmap Version: 1.0.12 in R language.

## 5. Conclusions

This study compared the morphological characteristics of mulberry fruits suitable for cultivation in tropical regions as two ripeness stages, and measured the physicochemical indicators of sugar, organic acids, and amino acid aroma substances. The main results are as follows: The full-ripe TLM was the longest, whereas the green-ripe BZM was the shortest. The full-ripe D10M had the largest fruit transverse stem, whereas the green-ripe BLM had the smallest. The single-fruit weight of green-ripe BZM was the smallest, whereas that of full-ripe D10M was the largest. The brix value of full-ripe TLM fruit was the highest. Fructose, glucose, and sucrose were the main sugars in all varieties, and fructose was the predominant sugar in the green-ripe mulberry varieties. However, in full-ripe mulberry varieties, sucrose was most common in BZM and BLM, whereas fructose was predominant in JL40M, D10M, and TLM. Oxalic acid, quinic acid, malic acid, and citric acid were the main organic acids in all varieties, and citric acid was the main organic acid in the BLM, TLM, D10M, and JL40M. Oxalic acid was the major component in the green-ripe BZM, whereas oxalic acid and malic acid were the most abundant in the full-ripe BZM. Some 20 free amino acids (Thr, Val, Met, Ile, Leu, Phe, Trp, Lys, His, Asp, Glu, Asn, Ser, Gln, Gly, GABA (γ- aminobutyric acid), Arg, Ala, Pro, Tyr, and Cys) were detected in all varieties, and the GABA content was highest in the full-ripe TLM sample, which is a key antihypertensive factor inhibiting angiotensin I-converting enzyme (ACE) activity in vitro. A total of 100 volatile compounds were identified in all the samples, including 31 esters, 20 aldehydes, 14 hydrocarbons, 11 alcohols, 10 acids, 6 ketones, and 8 others, the most abundant of which were aldehydes. This study provides a multilayered analysis and evaluation of the characteristics and quality of mulberry fruits.

## Figures and Tables

**Figure 1 plants-13-02763-f001:**
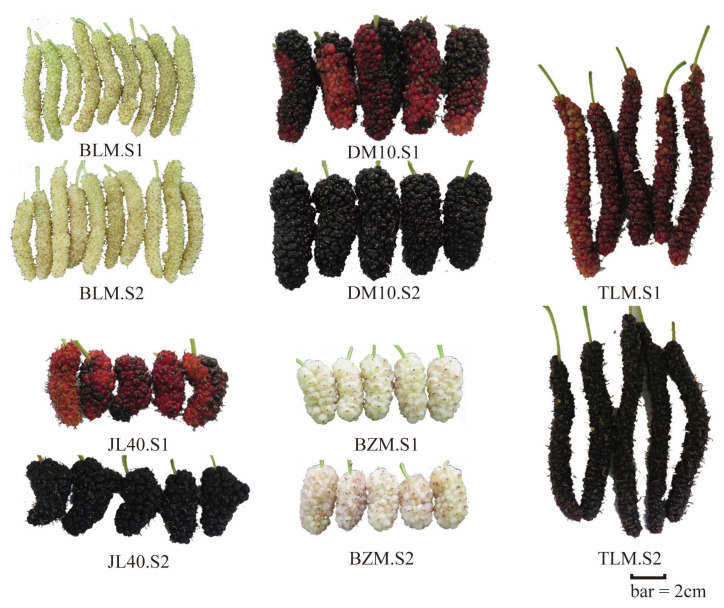
Five different varieties of mulberry with two ripeness stages.

**Figure 2 plants-13-02763-f002:**
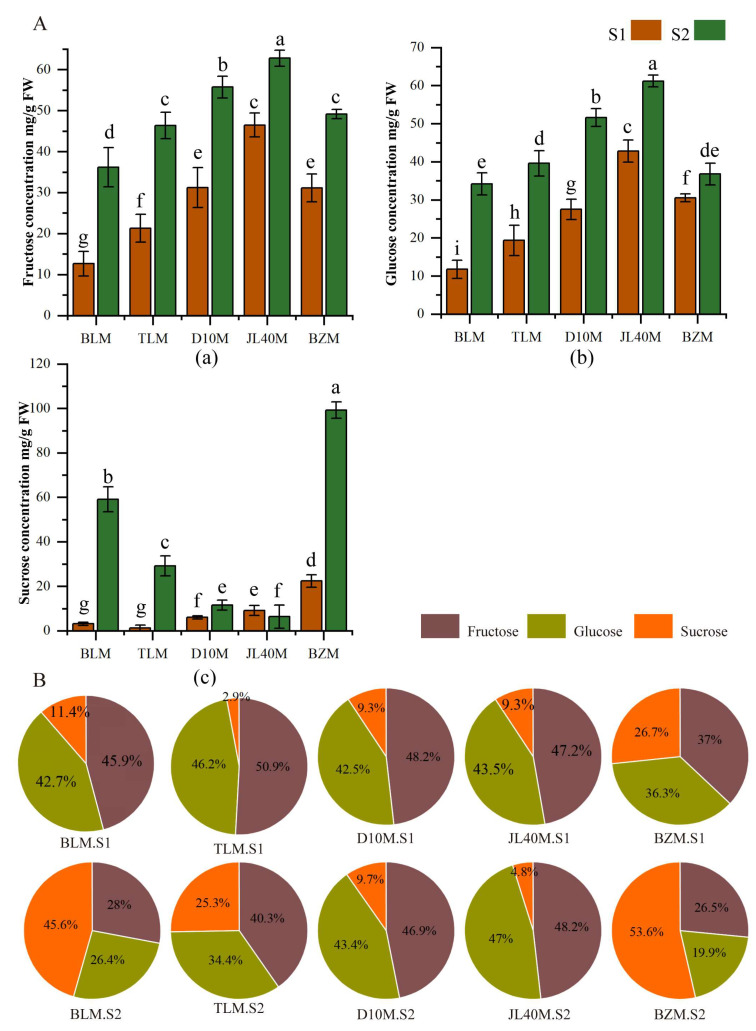
Sugar contents and the ratios of mulberry cultivars at different ripeness stages. (**A**) Changes in soluble sugar content: (a) Fructose concentration; (b) Glucose concentration; (c) Sucrose concentration, (**B**) The ratio of individual sugar. Different letters among the ten samples indicate significant differences according to one-way ANOVA and Duncan test with *p* ≤ 0.05.

**Figure 3 plants-13-02763-f003:**
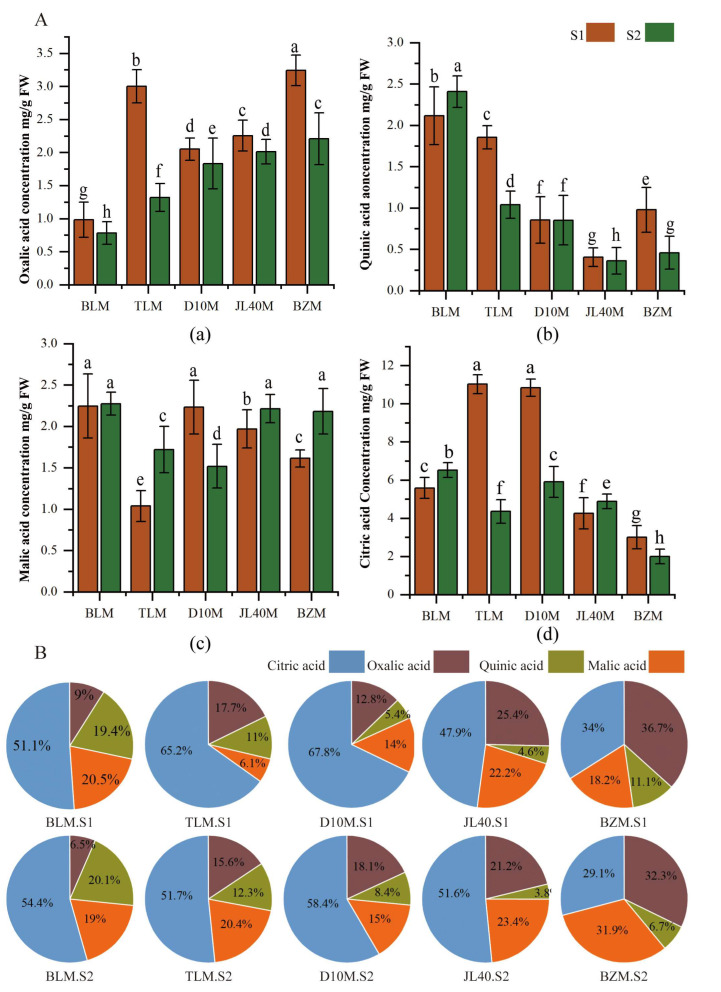
Organic acid content (mg/g fresh weight, FW) and ratio in mulberry cultivars at different ripeness stages. (**A**) Changes in organic acid content: (a) Oxalic acid concentration; (b) Quinic acid concentration; (c) Malic acid concentration; (d) Citric acid concentration. (**B**) The ratios of individual organic acids. Different letters among the ten samples indicate significant differences according to one-way ANOVA and Duncan test with *p* ≤ 0.05.

**Figure 4 plants-13-02763-f004:**
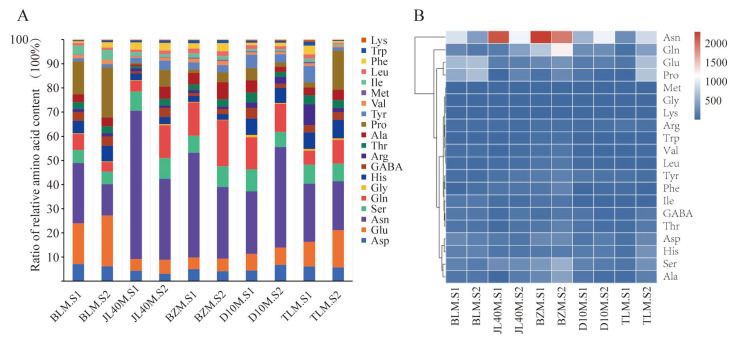
Amino acid contents in mulberry fruits during development and ripening. (**A**) Relative abundance of grouped volatiles in mulberry cultivars at different ripeness stages using SPME-GC-MS analysis. (**B**) Heat map of 20 amino acids in mulberry cultivars with different ripeness stages.

**Figure 5 plants-13-02763-f005:**
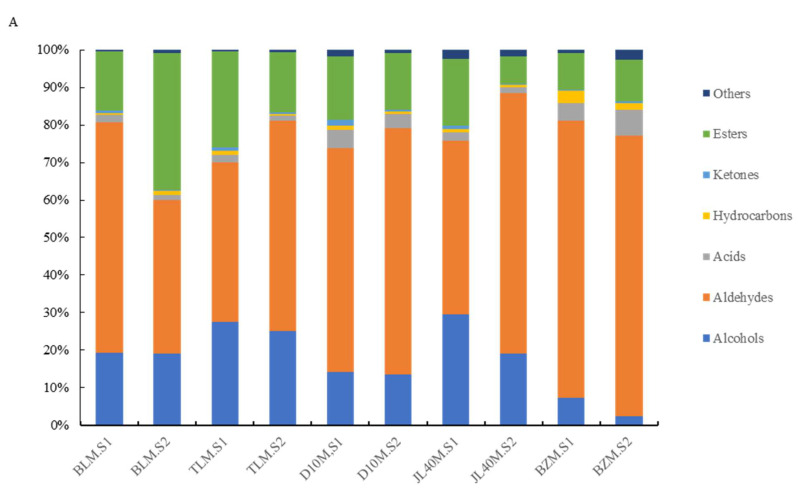
Volatile compound contents in mulberry fruits during development and ripening. (**A**) Relative abundance of grouped volatiles in mulberry cultivars at different ripeness stages using SPME-GC-MS analysis. (**B**) Heat map of 52 selected volatiles (peak area, filtration interquartile range threshold = 0.05%) in mulberry cultivars with different ripeness stages.

**Table 1 plants-13-02763-t001:** Mean values of fruit morphology and brix of mulberry cultivars.

Variety	Length (cm)	Transverse Stem (mm)	Weight (g)	Brix
D10M.S1	49.00 ± 0.58 b	19.00 ± 0.58 a	9.30 ± 0.15 b	11.10 ± 0.55 f
D10M.S2	49.33 ± 0.33 b	20.00 ± 0.57 a	9.63 ± 0.03 a	14.80 ± 0.15 d
JL40M.S1	30.33 ± 1.86 c	13.33 ± 0.67 bc	4.50 ± 0.06 f	10.07 ± 0.09 f
JL40M.S2	31.33 ± 1.20 c	14.33 ± 0.33 b	5.07 ± 0.09 e	13.57 ± 0.58 de
TLM.S1	82.00 ± 1.73 a	10.00 ± 0.58 d	6.90 ± 0.06 d	16.97 ± 0.55 c
TLM.S2	83.67 ± 0.88 a	10.33 ± 0.33 d	7.13 ± 0.03 c	25.90 ± 0.53 a
BLM.S1	46.00 ± 1.15 b	7.33 ± 0.33 e	3.70 ± 0.06 g	12.57 ± 0.33 e
BLM.S2	48.33 ± 1.20 b	8.33 ± 0.33 e	3.80 ± 0.06 g	22.47 ± 0.37 b
BZM.S1	23.67 ± 0.33 d	12.00 ± 0.58 c	2.27 ± 0.03 h	14.27 ± 0.15 d
BZM.S2	24.33 ± 0.33 d	13.33 ± 0.33 bc	2.37 ± 0.03 h	22.77 ± 0.58 b

Note: Different letters indicate significant differences between samples.

## Data Availability

Data are contained within the article.

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
