# Peer review of "Analysis of Main Components of Five Mulberry Varieties in Tropics"

_plants, 2024, doi:10.3390/plants13192763_

Round 1
Reviewer 1 Report
Comments and Suggestions for Authors
My comments:
1/ keywords must not overlap with the title; please let the authors know how keywords should be chosen
2/ in the introduction, please clearly state:
- research problem
- purpose of the research
- research hypothesis
In addition, please articulate the novelty of the work justifying the need for the research and its publication
3/ for all equipment used during the study, please use the notation: name (model, manufacturer, city, country)
4/ for analyses where standard curves are used, please add information: analyte concentration range, curve equation, fit
5/ for chromatographic analysis, please add chromatograms: standards and specific samples
6/ for chromatographic analyses, please specify for which compounds the standardization was performed, please provide the concentration ranges of standards, curve equations, fits; in addition, please provide LOD, LOQ
7/ statistics - why was parametric analysis performed for such a small number of samples? When the number of samples is less than 20, non-parametric analysis should be performed
8/ please provide better quality graphics, currently the quality is very poor
9/ please add doi numbers in references
10/ methodology - please provide FULL and DETAILED descriptions
11/ add informations about technological and analytical replications
Reviewer 2 Report
Comments and Suggestions for Authors
The work is undoubtedly important for understanding diversity of mulberry fruits chemical composition and its changes during ripening. The work is certainly interesting and has scientific novelty.
However, significant corrections should be done. The manuscript need to be improved according to Instructions for authors, especially references. English in the manuscript may be clearer. Sometimes there are absent spaces between words and dots at the end of sentences. Sometimes abbreviations are confused. Besides, abbreviations, specific terminology, chemical compounds names, and measuring units should be carefully checked because they are not identical throughout the text, tables and figures. For example, according to usage frequency by authors it is recommended to use the following phrases
- mulberry varieties (instead of mulberry species and even mulberries),
- green-ripe and full-ripe (instead of green ripe, green and full ripe, fullripe, ripe),
- ripeness stages (instead of ripeness, ripening, ripeness levels, stages of ripeness, maturity, maturation, mature stages and so on).
As for organic acids, are they as malate or as malic acid (line 129, for example)?
Abstract
For the first time give the full Latin name of mulberry. Besides, give abbreviations next to corresponding variety names.
In line 21 “than other disaccharides” should be changed as “than other sugars”.
Introduction
Please explain what is meant by the words “minerals in the human body were detected in the Chinese mulberry cultivar” (line 51).
In line 53 “physical and chemical characteristics” should be changed as “morphological and chemical characteristics of fruits”.
Please explain what is meant by the words “effective development and utilization” (line 60).
Results
In line 66 add that it was two-way ANOVA.
In line 67 it is given fruit stem length (and in line 277, too) and in Table 1 (and in line 79, too) it is given transverse stem. Please improve.
In line 73 add that it is shown in Table S1.
In Figure 1 use dots in abbreviations between mulberry variety and ripeness stage (as in Figure 2). JL40M should be instead of JL40.
In Table 1 omit the word body and explain what do different letters next to values indicate.
In lines 80 and 341 there should be BLM instead of BTM and in lines 81 and 342 there should be BZM instead of BTM.
In Figure 2A caption add information about graphs (a)-(c). In Figure 2A(a) the first column is without letter; please add. In Figure 2A(b) the fifth column should have the letter g instead of the letter j, should not it? In Figure 2A(c) the tenth column should have the letter a instead of the letter c, should not it? In Figure 2B please correct the legend. Besides, JL40M should be instead of JL40.
In line 118 add that it is shown in Table S2.
In lines 121 and 348 there should be JL40M instead of GL40M.
In line 127 there should be BLM instead of TLM.
In line 129 there should be TLM instead of BLM.
In Figure 3A caption add information about graphs (a)-(d). In Figure 3A(d) the letter d is absent. Please improve. In Figure 3B there should be JL40M instead of JL40.
How many amino acids were found in all tested mulberry varieties (lines 140-146, Figure 4, Table S3)? Cys is listed in the text (line 144) but in Figure 4 and Table S3 it is absent. Please improve.
In line 142 add that it is shown in Table S3.
Check the information given in lines 150-156. It seems that something was omitted.
In line 151 there should be JL40M (or, may be, JL40M.S1) instead of JM40.
In line 155 there should be D10M.S1 instead of DM10S1.
In line 160 “which ncreased during” should be changed as “which increased during”.
In line 166 “Amino compound content” should be changed as “Amino acid content”.
In the section 2.4 add that it is shown in Table S4.
Rephrase the words “The common aroma component species” (lines 179-180).
In lines 187, 195 and 201 there should be JL40M instead of JL40.
According to the text, content of 2-hexen-1-ol, acetate, (E)- is 8.75% (line 193) and according to Table S4, it is 8.65%. Please improve. In which mulberry variety it amounts to 5.15%?
In line 199 please correct YLM.
In line 200 there should be JL40M instead of JLM40.
Discussion
The Discussion section should be revised and analysis of studied metabolites should be given. In fact, it should not repeat the Results section. Now there is no analysis of aroma components and odor, of differences in compounds spectra between two ripeness stages and of tropical climate conditions influence on compounds spectra as it is pointed in the manuscript’s title and in Abstract. Some of the compounds were earlier analyzed, for example, in [15], and this reference may be addressed not only in Introduction section, but in Discussion section, too. How fruits’ morphology, ripeness stage, odor, and chemical composition may be connected? Are there any compounds, found by the authors of the submitted article, which had not been described in mulberry before? A comparison with other fruits may be provided.
Rephrase the words “did not contain the most fructose” (line 245).
Rephrase the words “GABA was the most abundant substances” (lines 263-265).
Materials and Methods
In the section 4.1 give information about S1 and S2 ripeness stages and the aforementioned tropical climate conditions.
What are the differences between BLM and TLM (lines 272-273)?
In line 273 there should be JL40M instead of JLM.
What was particle size (lines 293, 296, and 307)? What was injection volume (lines 293, 296, and 311)?
In line 301 “Hcl was added” should be changed as “HCl was added”.
Rephrase the words “The column length was 4.6 mm × 60 mm” (line 307).
Which buffer was used for amino acid analysis (line 308)?
In the section 4.5 please name all the measurements done for this very manuscript and add information about two-way ANOVA.
Was P < 0.05 (line 336) or P ≤ 0.05 (lines 108 and 137)?
Conclusions
In lines 338-339 rephrase the sentence.
Rephrase the words “the content of GABA was found most” (line 352).
Appendix A
In Table S2 “total rganic acid” should be changed as “total organic acids”.
In Table S3 add measuring units and explain what do different letters next to values indicate. Is Pro content in BLM.S1 the same as in BZM.S2? For Val content, the letter f should be instead of the letter g, should not it? For Phe content, the letter g is absent. Please improve.
Comments on the Quality of English LanguageIt may be improved.
Reviewer 3 Report
Comments and Suggestions for Authors
The study compared the morphology and chemical properties of berries of five mulberry varieties at two ripeness levels.
The purpose and justification of the work are good. The materials and methods section sounds good. The statistics are robust and correctly applied. The results, discussion and conclusions are appropriate, but require some correction. In my opinion, the work can be published after corrections.
Line 46: add reference number.
Line 54: Two-stage of maturity instead of two harvest period from
Tables and Figures
Explain the abbreviations of a variety names and degrees of maturity in the description of Tables and Figures.
Other abbreviations are used in Tables and Figures, e.g. Table 1 - D10M but in Figure 1 - DM10.
(the same BLM – Table 1 and BTM – line 80; BZM – Table 1 and BTM – line 81).
Figure 1 – BLMS1 but Figure 2B – BLM.S1
Standard deviations given in Tables should have the same number of decimal places as the values ​​provided.
When discussing the results and quoting numerical values ​​in the text of the paper, there is no need to provide standard deviations included in the Tables.
Figure 2: Axis titles: concentration instead of Concentration,
Figure B legend: Sucrose or Sructose
Line 118: oxalate and citrate.
Line 147: Typically, the nutritional value of products is expressed per 100 g.
Lines 150-152: The sentence is incomprehensible and requires rephrasing. “The highest proportion of Asn is in JM40 during the S1 stage, except for the D10M variety, which decreases with increasing maturity; “
Lines 152-156: In my opinion, a trend of increase or decrease in the content of ingredients could be concluded if several stages of maturity were examined. When comparing two stages of ripeness, it is only possible to determine a higher or lower content of ingredients. Especially when the content increased in some varieties and decreased in others.
Line 155: DM10S1 is a sample not a stage
Line 166: mulberry (or fruit) instead fruit tree
Line 168: amino acid instead of amino
Figure 5B is illegible.
Lines 189-190: Check carefully the sentence. Is incomprehensible and incorrect. I propose: The content of esters in mulberry fruits was usually higher in the green ripe stage than in full-ripe ones, only in the BLM variety it was lower in full-ripe ones, up to 34.77%.
Line 199: YLM?
Line 236: fruit No.1?
Line 243: [22] instead of (2012)
Line 288: Does this mean that the entire 50 mL was filtered through a 0.45 μm Millipore filter?
Line 339: replace detected by examined
Table S2: total organic acid instead of total rganic acid
References [21]: J. Berry Res.
For more see attached file.

Round 2
Reviewer 1 Report
Comments and Suggestions for Authors
Accept in current form
Author Response
Dear Academic Editor:
Thanks for the comments on my manuscript " Analysis of main components of five mulberries in tropics" (plants-3116229). We appreciate and accept the modification suggestions and have revised the manuscript strictly according to the attached Reviewers' comments. The detailed responses to the reviewers’ comments are presented as follows:
- The identification of volatile compounds, should be done perfectly using two methods of detection (MS and RI), and in both cases you have to supply the KI values, I don't see any problem, to report the KI values, this is a basic practice for the researchers using this technique, and is not related to the absolute content as mentioned by the authors (it has nothing to do with the content, it's for identification), MS fingerprint is not enough.
Response 1: Thank you very much for your feedback, we have added the KI value of compound in Table S5
2- Again, the table (S4), contains unnatural compounds, please recheck again after adding the KI values.
Response 2: We have rechecked the composition of the compounds and believe that certain substances, although possibly synthesized chemically, exist in nature. Additionally, since some substances may be intermediate products in the metabolism of mulberries, we have not deleted them. We have modified the corresponding data in the original text.
- the paragraph 4.3, again needs careful revision, the title has nothing to do with the content, or vice versa.
Response 3: We have rechecked the content of section 4.3 and have made revisions. If there are any imperfections, please criticize and correct us. Thank you again for your suggestions on the shortcomings of our article.
